

# A hybrid blockchain-based solution for secure sharing of electronic medical record data

Gang Han[1,2,3], Yan Ma[3], Zhongliang Zhang[1] and Yuxin Wang[3]

[1] School of Management, Hangzhou Dianzi University, Hangzhou, Zhejiang, China
[2] The State Key Laboratory of Integrated Service Networks, Xidian University, Xi'an, Shaanxi, China
[3] The School of Cyberspace Security, Xi'an University of Posts and Telecommunications, Xi'an, Shaanxi, China

## ABSTRACT

Patient privacy data security is a pivotal area of research within the burgeoning field of smart healthcare. This study proposes an innovative hybrid blockchain-based framework for the secure sharing of electronic medical record (EMR) data. Unlike traditional privacy protection schemes, our approach employs a novel tripartite blockchain architecture that segregates healthcare data across distinct blockchains for patients and healthcare providers while introducing a separate social blockchain to enable privacy-preserving data sharing with authorized external entities. This structure enhances both security and transparency while fostering collaborative efforts across different stakeholders. To address the inherent complexity of managing multiple blockchains, a unique cross-chain signature algorithm is introduced, based on the Boneh-Lynn-Shacham (BLS) signature aggregation technique. This algorithm not only streamlines the signature process across chains but also strengthens system security and optimizes storage efficiency, addressing a key challenge in multi-chain systems. Additionally, our external sharing algorithm resolves the prevalent issue of medical data silos by facilitating better data categorization and enabling selective, secure external sharing through the social blockchain. Security analyses and experimental results demonstrate that the proposed scheme offers superior security, storage optimization, and flexibility compared to existing solutions, making it a robust choice for safeguarding patient data in smart healthcare environments.

## INTRODUCTION

Blockchain technology, recognized as a decentralized and secure distributed ledger system, has been significantly adopted in sectors such as finance and supply chains (*Casino, Dasaklis & Patsakis, 2019*; *Gonczol et al., 2020*; *Powell et al., 2021*). Its decentralized nature ensures that there is no single point of failure, making it resilient against attacks and system breakdowns, while its inherent immutability guarantees that once data is recorded, it cannot be tampered with. These characteristics are particularly valuable in the healthcare domain, where the integrity and trustworthiness of sensitive medical records are paramount. Moreover, blockchain's transparency allows authorized healthcare providers, patients, and other entities to securely verify and access data, which enhances trust in the

Corresponding author
Yan Ma, vickiem528@gmail.com

system while safeguarding patient privacy through cryptographic techniques. Given these benefits, there has been burgeoning interest in applying blockchain technology to healthcare in recent years, where it has been envisioned as a novel means to manage patient records, diagnostic data, prescriptions, and other sensitive information (*Stafford & Treiblmaier, 2020*; *Hang, Choi & Kim, 2019*; *Chamola et al., 2022*).

Most previous studies on blockchain-based healthcare systems adopted an on-chain and off-chain storage model to achieve system decentralization and alleviate storage pressure (*Miyachi & Mackey, 2021*; *Kumar, Marchang & Tripathi, 2020*). For instance, vast amounts of encrypted data are stored on cloud servers, while single blockchains store the addresses. However, in practical scenarios, this storage model can lead to significant disarray in the storage spaces both on-chain and off-chain, consequently reducing system functionality and increasing the likelihood of system attacks.

Some current solutions improve data accessibility among healthcare providers by modifying access control policies or managing workflows (*Tanwar, Parekh & Evans, 2020*; *Khatoon, 2020*; *David et al., 2023*). These approaches typically use a single blockchain to store all medical-related data, which enhances system security and logical coherence to some extent but does not fundamentally solve the issue of chaotic storage models. In terms of system security, some methods alter the blockchain consensus mechanism (*Garcia, Ramachandran & Ueyama, 2022*; *Ali et al., 2023*; *Gupta et al., 2019*) to avoid 51% attacks. However, due to the sensitivity and integrity requirements of medical data, more decentralized and verifiable consensus methods, which still carry certain risks of attacks, are necessary (*Košťál et al., 2018*).

To address these issues, this article proposes a novel method that introduces a hybrid blockchain-based solution for the secure sharing of electronic medical record (EMR) data. This approach employs three blockchains to identify different functions within the healthcare system and designs a unique cross-chain signature algorithm tailored for this system. This method enhances data security while optimizing storage space. Our approach makes the system resilient to 51% attacks, achieves data fitting, and improves the logical structure of data storage.

The contributions of this article are as follows:

- A hybrid blockchain-based solution is proposed for secure sharing of EMR data, which rationally distributes healthcare system functions through a multi-chain storage model. This model reduces storage pressure and enhances the system's resistance to 51% attacks.
- A cross-chain signature algorithm is designed to improve data privacy protection, achieve data fitting, and alleviate the chaotic storage space issue in blockchain healthcare systems, making the storage model more organized.
- During external data sharing, the social blockchain is used to record transactions, while the implementation of an efficient data layering strategy ensures secure and controlled access based on sensitivity levels.

The remainder of this article is organized as follows: "Related Work" discusses related work, "Preliminaries" introduces relevant background knowledge, "Hybrid Blockchain-Based Solution for Secure Sharing of EMRs" presents the proposed system solution, "Experiments and Analysis" provides experiments and security analysis, and finally, "Discussion" analyzes the limitations of the solution and concludes the article.

## RELATED WORK

Blockchain technology has garnered significant attention in the design of EMR systems to address the challenges of fragmented health data, privacy protection, and secure data sharing. Previous studies typically adopt a hybrid on-chain and off-chain storage model to achieve decentralization and alleviate storage pressure. For instance, a distributed electronic health record (EHR) ecosystem was proposed that integrates EMR into a private and permissioned blockchain. This approach aims to unify fragmented patient records across various healthcare organizations, enhancing data consistency and security (*Cerchione et al., 2023*). Similarly, *Chelladurai & Pandian (2022)* proposed a blockchain-based EHR system that offers a regulated solution for patients, physicians, and healthcare providers that addresses data fragmentation issues. *Kim et al. (2020)* introduced a secure and efficient solution for managing EHRs using blockchain for data integrity, access control, and secure health data sharing, combined with cloud computing. *Fatokun, Nag & Sharma (2021)* further expanded on this concept by proposing a patient-centric EHR system on the Ethereum blockchain platform that provides patients with greater control over their data and eliminates the need for third-party systems.

Other researchers have focused on enhancing data privacy and system scalability. *Shuaib et al. (2022)* proposed a blockchain-based healthcare data-sharing system that integrates a decentralized file system and a threshold signature to mitigate privacy-linking attacks and scalability challenges. *Liu et al. (2020)* addressed secure storage and sharing of EMRs with a consortium blockchain-based solution that incorporates anonymous and traceable identity privacy protection, dual blockchain and cloud server storage, and an improved proxy re-encryption scheme. In addition, *Guo et al. (2022)* developed a hybrid blockchain-edge architecture employing attribute-based cryptographic mechanisms for managing EHRs. This architecture features an innovative attribute-based signature aggregation (ABSA) scheme, multi-authority attribute-based encryption (MA-ABE), and Paillier homomorphic encryption (HE) for patient anonymity and EHR security (*Guo et al., 2022*). *Liu et al. (2022)* suggested using proxy re-encryption and sequential multi-signature combined with cloud platform services to further protect patient privacy data on the blockchain. *Yuan et al. (2022)* proposed a detailed, secure sharing scheme for medical data leveraging blockchain technology, addressing the issues of low throughput and instability in single-chain models while enhancing data confidentiality.

Recent studies have also explored the use of Byzantine consensus mechanisms in blockchain-based healthcare systems. For example, a blockchain-based healthcare platform with Byzantine fault tolerance (BFT) was proposed, ensuring data integrity,

**Table 1 Summary of related work.**

| Paper | Timeline | Blockchain type | Consensus algorithm | Data location | Off-chain storage | Fine-grained permission | Performance eval | Cross-chain signature | Data shared externally | Data storage performance |
|---|---|---|---|---|---|---|---|---|---|---|
| *Cerchione et al. (2023)* | 2023 | – | – | On-Chain | – | ✗ | √ | ✗ | ✗ | + |
| *Chelladurai & Pandian (2022)* | 2022 | – | – | On-Chain, Off-Chain | Cloud storage | ✗ | ✗ | ✗ | ✗ | + |
| *Kim et al. (2020)* | 2020 | – | PBFT | On-Chain, Off-Chain | Cloud storage | ✗ | √ | ✗ | ✗ | + |
| *Fatokun, Nag & Sharma (2021)* | 2021 | Ethereum | – | On-Chain, Off-Chain | Cloud storage | √ | √ | ✗ | ✗ | ++ |
| *Shuaib et al. (2022)* | 2022 | Ethereum | PBFT | On-Chain, Off-Chain | Cloud storage | √ | √ | ✗ | √ | ++ |
| *Liu et al. (2020)* | 2021 | – | – | On-Chain, Off-Chain | Cloud storage | ✗ | ✗ | ✗ | √ | + |
| *Guo et al. (2022)* | 2022 | Hyperledger Caliper | – | On-Chain, Off-Chain | IPFS | ✗ | √ | ✗ | ✗ | ++ |
| *Liu et al. (2022)* | 2022 | – | POS | On-Chain, Off-Chain | Cloud storage | ✗ | √ | ✗ | √ | ++ |
| *Yuan et al. (2022)* | 2022 | – | POW | On-Chain, Off-Chain | IPFS | √ | √ | ✗ | ✗ | ++ |
| *Okegbile, Cai & Alfa (2022)* | 2022 | – | PBFT | On-Chain, Off-Chain | Cloud storage | ✗ | √ | ✗ | ✗ | + |
| *Zaabar et al. (2021)* | 2021 | Hyperledger Caliper | PBFT | On-Chain, Off-Chain | Cloud storage | ✗ | √ | ✗ | √ | ++ |
| *Hegde & Maddikunta (2023)* | 2023 | Hyperledger Caliper | PBFT | On-Chain, Off-Chain | IPFS | ✗ | √ | ✗ | ✗ | ++ |
| Our solution | – | Ethereum | POS | On-Chain, Off-Chain | IPFS | √ | √ | √ | √ | +++ |

confidentiality, and availability, which is crucial for healthcare applications (*Okegbile, Cai & Alfa, 2022*). Another study introduced an efficient and secure health data sharing framework using blockchain with Byzantine consensus, which addressed issues like data tampering and unauthorized access (*Zaabar et al., 2021*). Additionally, a novel approach for secure EHR using Hyperledger Fabric with BFT was explored that would enhance patient data security and accessibility while maintaining high performance and reliability (*Hegde & Maddikunta, 2023*).

As shown in Table 1, while these initiatives illustrate significant progress in integrating blockchain technology into EMR systems, they often rely on single blockchain models or hybrid storage solutions that may lead to chaotic storage management and reduced system functionality. Additionally, current methods for enhancing system security, such as modifying consensus mechanisms, still face challenges in completely mitigating the risks of attacks, especially given the sensitivity of medical data. To address these limitations, our research introduces a novel hybrid blockchain-based solution for the secure sharing of EMR data. By employing three blockchains for different functions within the healthcare system and designing a unique cross-chain signature algorithm, our approach optimizes storage space, enhances data security, and improves system resilience to 51% attacks. This

method ensures organized data storage, provides a more robust solution for secure EMR data sharing, and advances the state of blockchain applications in healthcare.

## PRELIMINARIES

This section provides a brief review of relevant knowledge.

### Blockchain-related theory

Blockchain represents a novel application paradigm of computer technology that integrates various cutting-edge technologies including distributed data storage, peer-to-peer (P2P) transmission, consensus mechanism, and encryption algorithms. It serves as a decentralized and trustless infrastructure that operates on a distributed computing paradigm. The theoretical foundations of blockchain-primarily draw upon information asymmetry theory, free currency theory, and Byzantine fault tolerance (BFT) theory, while the technical support is provided by P2P network technology, timestamp technology, asymmetric encryption, smart contracts, and database technology (*Zheng & Lu, 2022*; *Xiong et al., 2022*; *Nakamoto, 2008*). Generally, the infrastructure of blockchain is comprised of a data layer, network layer, consensus layer, incentive layer, contract layer, and application layer, as illustrated in Fig. 1.

### Attribute-based encryption

The basic idea of attribute-based encryption (ABE) is to integrate the access control of data into the decryption process of the cipher text, providing a new perspective on the access control of encrypted data (*Zhang et al., 2020*; *Rasori et al., 2022*). The most important feature of this encryption method is that it does not rely on the user's identity information to encrypt and decrypt the data, but on a set of attributes of the user, and only when the user's attributes satisfy the access policy defined in the ciphertext can the user successfully decrypt the original text.

There are two main types of attribute-based encryption techniques, key-policy attribute-based encryption (KP-ABE) and ciphertext-policy attribute-based encryption (CP-ABE) (*Al-Dahhan et al., 2019*). In KP-ABE, the key is determined by an access structure and the ciphertext is marked by a set of attributes. A user can only decrypt a ciphertext if the access structure of his key matches the set of attributes of the ciphertext. In contrast, in CP-ABE, the access policy is specified by the ciphertext and the user's key is marked by a set of attributes. The user can only decrypt a ciphertext if the set of attributes of the key satisfies the access policy of the ciphertext.

### BLS signature

The Boneh-Lynn-Shacham (BLS) signature is a type of digital signature scheme that offers a short, computationally efficient signature and the ability to aggregate signatures (*Boneh, Lynn & Shacham, 2001*). The BLS signature scheme is based on bilinear pairings on elliptic curves, which makes it possible to compress multiple signatures from multiple users into a single signature. This aggregation capability is particularly valuable for systems that need

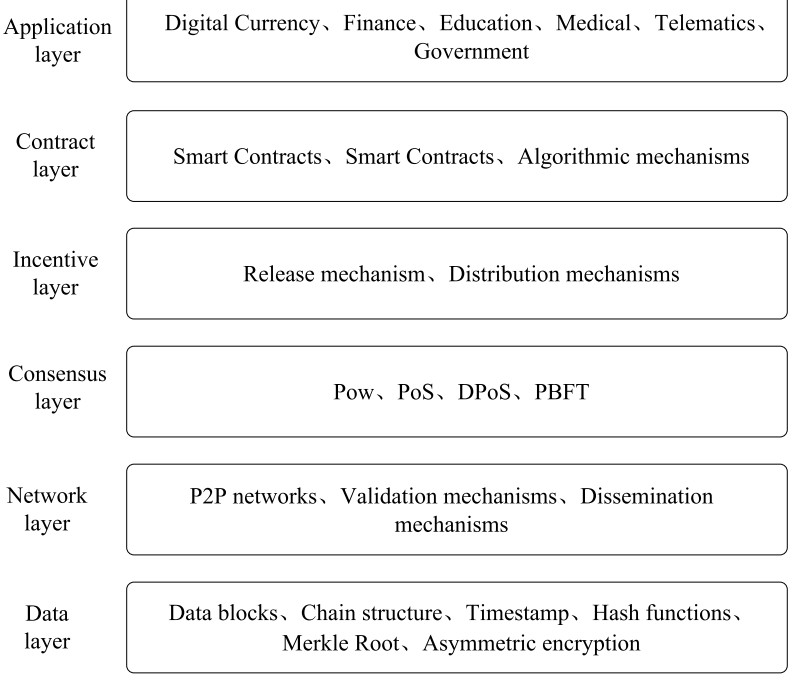

**Figure 1** **Blockchain infrastructure diagram.**

to manage a large number of signatures, like blockchain networks. This algorithm needs a bilinear pairing $\mathbb{G}_0 \times \mathbb{G}_1 \rightarrow \mathbb{G}T$. A bilinear pairing satisfies the following properties:

Bilinearity: $e(g_0^a, g_1^b) = e(g_0, g_1)^{ab}, \forall a, b \in Z_q$

Non-degeneracy: $e(g_0, g_1) \neq 1$

Efficient Computability: The map $e$ can be efficiently computed.

All three groups $(\mathbb{G}_0, \mathbb{G}_1, \mathbb{G}_T)$ have prime order $q$. Let $g_0$ and $g_1$ be generators of $\mathbb{G}_0$ and $\mathbb{G}_1$ respectively. The scheme also requires a hash function $H_0: \{0, 1\}^* \rightarrow \mathbb{G}_0$.

BLS signature aggregation works as follows:

KeyGen (): Choose a random $\alpha \xleftarrow{R} \mathbb{Z}_q$ and $h \leftarrow g_1^a \in \mathbb{G}_1$. Output $pk := (h)$ and $sk := (\alpha)$.

Sign $(sk, m)$: For a message $m$, compute the signature $\sigma \leftarrow H_0(m)^\alpha \in \mathbb{G}_0$. The signature is a single group element.

Verify $(pk, m, \sigma)$: Given $pk, m$, and $\sigma$ verify the signature by checking $e(g_1, \sigma) = e(pk, H_0(m))$. If the equality holds, output "accept"; otherwise, output "reject".

Signature Aggregation: For a given set of triples $(pk_i, m_i, \sigma_i)$ where $i = 1, 2, \ldots \ldots, n$, it is possible to combine the individual signatures $\sigma_1, \sigma_2, \sigma_n \in \mathbb{G}_0$. The correctness of the aggregated signature $\sigma$ can be verified by ensuring that the following condition hold:

$$e(g_1, \sigma) = e(pk_1, H_0(m_1)) \cdot e(pk_2, H_0(m_2)) \ldots \ldots e(pk_n, H_0(m_n))$$

In the special case where all messages are identical, the verification equation simplifies significantly. Instead of $n$ pairings, only two pairings are needed for validation:

$$e(g_1, \sigma) = e(pk_1 \cdot pk_2 \cdot \ldots \ldots \cdot pk_n, H_0(m_1))$$

**Table 2 List of symbols and notations.**

| Symbol | Description |
| --- | --- |
| $\mathbb{G}_0$ | Bilinear pairing group 0 |
| $\mathbb{G}_1$ | Bilinear pairing group 1 |
| $\mathbb{G}_T$ | Target group for bilinear pairing |
| $q$ | Prime order of the groups |
| $g_0$ | Generator of $\mathbb{G}_0$ |
| $g_1$ | Generator of $\mathbb{G}_1$ |
| $H_0$ | Hash function treated as a random oracle |
| $H_1$ | Second hash function $\mathbb{G}_1^n \rightarrow R^n$ |
| $sk$ | Secret key |
| $pk$ | Public key |
| $mk$ | Master key for CP-ABE |
| $M$ | Message |
| $e$ | Bilinear pairing function $e\colon \mathbb{G}_0 \times \mathbb{G}_1 \rightarrow \mathbb{G}_T$ |
| $PI_i$ | Personal information (name, age, gender, *etc.*) |
| $C_i$ | Ciphertext of medical data |
| $H(C_i)$ | Hash value of ciphertext $C_i$ |
| $k$ | Symmetric key |
| $E_{PK_u}(k)$ | Symmetric key $k$ encrypted with the patient's public key |
| $D_i$ | Dataset including $D_i = C_i||H_i(C_i)||E_{PK_u}(k)$ |
| $S_A$ | Aggregate signature |
| $S_{A_s}$ | Aggregate signature for enhancing data privacy |
| $S_{A_f}$ | Aggregate signature for achieving data fitting |
| $T_i$ | Time at which a user uploads data |
| $A$ | Attribute set |
| $\mathcal{H}$ | Signature sets |

Our scheme is built on the foundation of the BLS signature algorithm, leveraging its aggregation capabilities and introducing optimizations to better suit the needs of blockchain-based medical data sharing systems.

## Notation table

To facilitate understanding of the proposed method, Table 2 summarizes the symbols used throughout this article.

# HYBRID BLOCKCHAIN-BASED SOLUTION FOR SECURE SHARING OF EMRS

## System architecture

The proposed scheme consists of three interconnected blockchains: the patient blockchain, healthcare provider blockchain, and social blockchain. A combination of on-chain and off-chain structures is employed to store medical records, providing the necessary flexibility and scalability to securely store and manage large volumes of sensitive healthcare

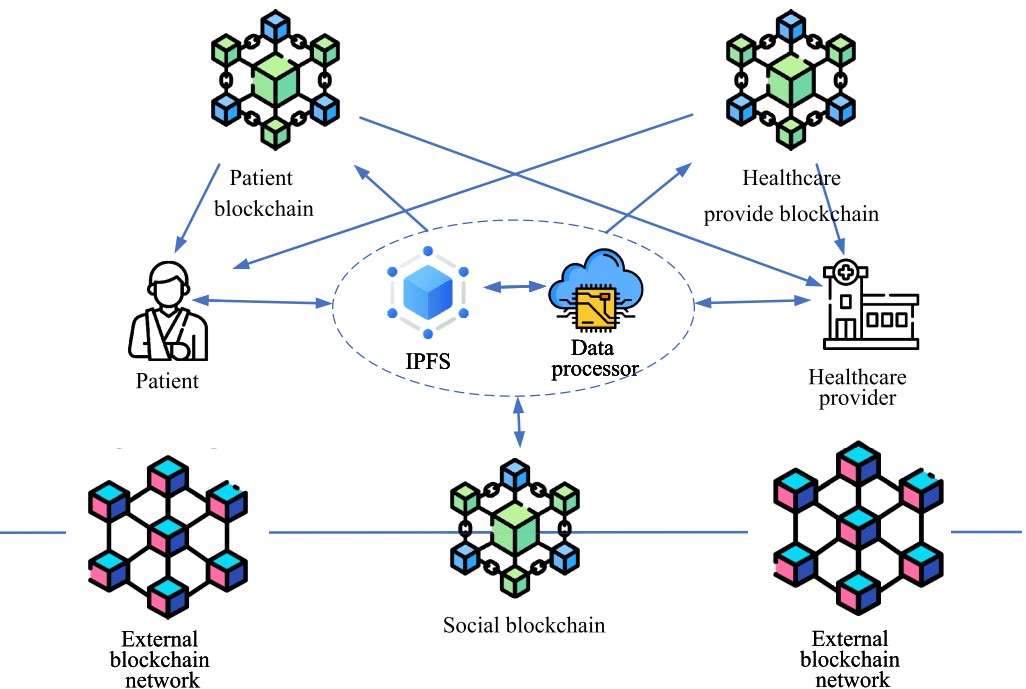

**Figure 2 Blockchain-based healthcare architecture.** Patient blockchain, Healthcare provider blockchain, Social blockchain, External blockchain network, Data Processor, IPFS, Patient, Healthcare provider: icons from Flaticon.com.

information. The blockchain-based healthcare architecture is shown in Fig. 2. The data sharing process of the system is illustrated in Fig. 3, which demonstrates the data flow between blockchains, the signature algorithm, and other steps.

### Patient blockchain

The patient blockchain stores the signature signed by the patient, along with hashed data that includes encrypted personal information $C_{pk}(PI_i)$. This information encompasses sensitive patient data such as name, age, gender, and other privacy-related information. To ensure the security and privacy of the data, it is uploaded to InterPlanetary File System (IPFS) by the patient and then hashed by IPFS. Given the smaller size of personal data compared to medical data, CP-ABE is employed to encrypt personal data, allowing for a higher degree of privacy protection. Personal information and EHR are stored separately in distinct blockchains to enhance the security of EHR. This approach prevents adversaries from associating medical data with specific patients, thereby reducing the risk of data breaches and safeguarding patient privacy.

Patients and healthcare providers can access data from the patient blockchain and retrieve ciphertext from IPFS. This ciphertext can then be decrypted to reveal the actual personal information of the patient. In typical scenarios, healthcare providers require access to medical data from a healthcare provider blockchain that is connected to a patient's personal information, which can then be used to make a diagnosis.

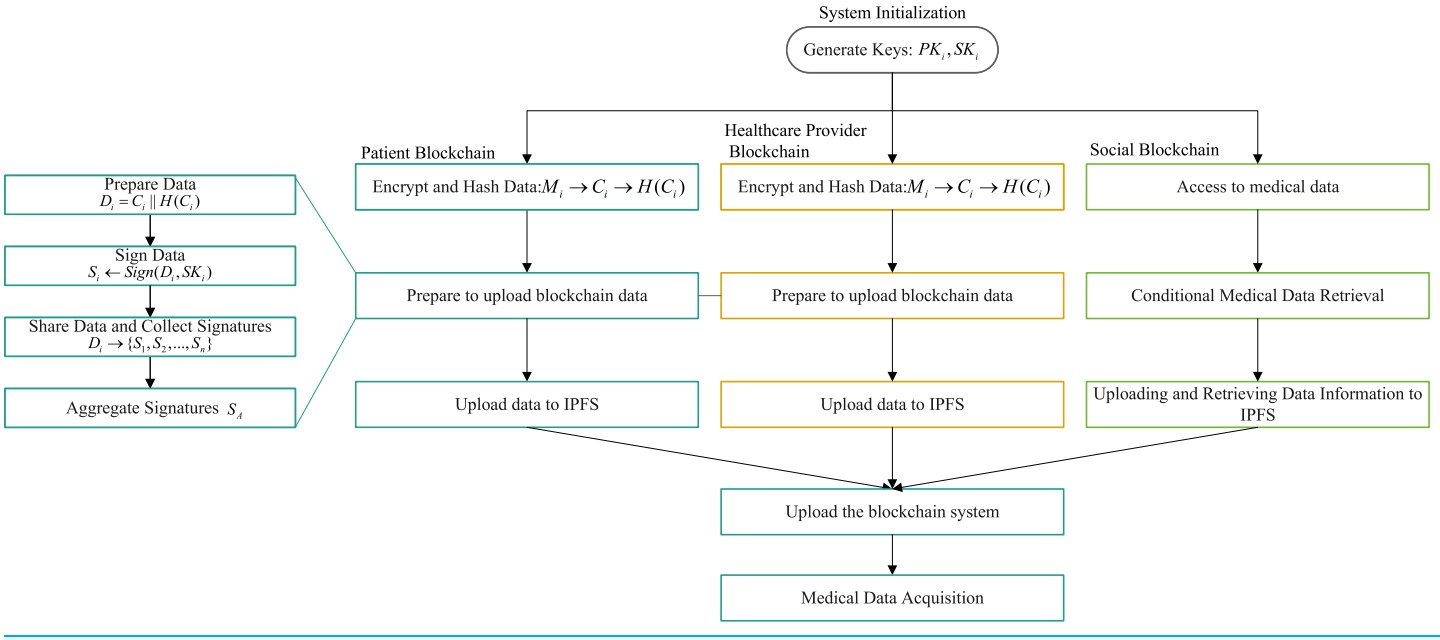

**Figure 3 Hybrid blockchain-based EMR data sharing flow.**

### Healthcare provider blockchain

The healthcare provider blockchain primarily stores aggregate signatures $S_A$, which are composed of signatures generated by various institutions, along with dataset $D_i$, where $T_i = D_i||S_A$. The dataset is comprised of the medical data ciphertext $C_i = E_k(M_i)$, its hash value $H(C_i)$, and the symmetric key $k$ encrypted with the patient's public key for encrypting medical data $E_{PK_u}(k)$. Specifically, $D_i$ can be expressed as $D_i = C_i||H(C_i)||E_{PK_u}(k)$. This approach ensures that the medical data and associated signatures are securely stored, while also maintaining the privacy and confidentiality of patient information.

Similarly to the patient blockchain, patients and healthcare providers can access data from the healthcare provider blockchain and retrieve ciphertext from IPFS. Given the critical nature of medical data during physician diagnostic and data access procedures, additional security measures are necessary to enhance the protection of sensitive medical data.

### Social blockchain

The social blockchain serves as a crucial component in our scheme, facilitating connections to external blockchain networks. Transactions involving data sharing with other systems are uploaded to this blockchain. Each transaction includes the hashed ciphertext that has already been shared with others and the signature of the data's owner. To enable better differentiation of which parts of a patient's EHR are shared, a data processor is used to classify the medical records, dividing the data into finer-grained

categories. When a patient transfers to another hospital, the social blockchain connects to the external blockchain network system, and the relevant patient data is transferred accordingly. The social blockchain does not require direct interaction with patients and healthcare providers. Our data classification scheme provides an effective solution for transferring different types of data, thereby reducing the workload for users.

## Cross-chain signature algorithm

This article proposes the cross-chain signature algorithm, built upon the BLS signature algorithm, which has been optimized and adapted to our proposed three-chain model. This adaptation enables efficient cross-chain user signature aggregation and data fitting. Our signature algorithm ultimately generates two types of aggregate signatures $S_{A_s}$, which enhances data privacy, and $S_{A_f}$, which achieves data fitting. The algorithm involves two primary user groups: patients within the patient blockchain and medical service providers within the medical service provider blockchain.

Regarding the aggregate signature $S_{A_s}$, it is assumed that when patients generate data, they transmit the ciphertext of the data along with other relevant information to all medical institutions. Subsequently, each user signs the ciphertext data. After obtaining the individual signatures from each user, an aggregate operation is performed to compute the aggregate signature $S_{A_s}$. Similarly, when new medical data is generated within medical institutions, the ciphertext is shared with other users, and the same signing and aggregation process is executed. For the aggregate signature $S_{A_f}$, users participating in the system continuously generate new data. Any user is required to compute the aggregate signature $S_{A_f}$ of all signatures $S_i$ generated before time $T_i$. Finally, at time $T_i$, users upload both $\left(S_{A_s}, S_{A_f}\right)$ to the new block.

Since each user has access to the same data ciphertext and signatures, attackers cannot identify the true source of the data, thereby preventing targeted attacks and enhancing the privacy of medical data. Additionally, the presence of the aggregate signature $S_{A_f}$ allows for the identification of all data signatures generated by a particular user, thereby achieving the fitting of heterogeneous data.

### *Aggregate signature $S_{A_s}$*

The process of obtaining the signature $S_{A_s}$ is illustrated in Fig. 4A, with the protocol flow shown in Fig. 4B. As an example of patient-generated data, the details of the signature process are as follows:

Our scheme needs a bilinear pairing e: $\mathbb{G}_0 \times \mathbb{G}_1 \to \mathbb{G}_T$, the hash function $H_0: \mu \to \mathbb{G}_0$, and a second hash function $H_1: \mathbb{G}_1^n \to R^n$ where $R := \{1, 2, \ldots, 2^{128}\}$.

a) KeyGen ( )

The system assigns the public key $PK_1$ and secret key $SK_1$ to medical staff for signing.

b) Prepare data $D_1$

The patient's personal data, denoted as $M_1$, is self-generated by the patient, followed by encryption to derive the ciphertext $C_1 = E_k(M_1)$. Subsequently, $C_1$ is subjected to a hashing process to yield the hashed data, denoted as $H(C_1)$. The culmination of this process results in the final prepared data $D_1 = C_1 \| H(C_1)$.

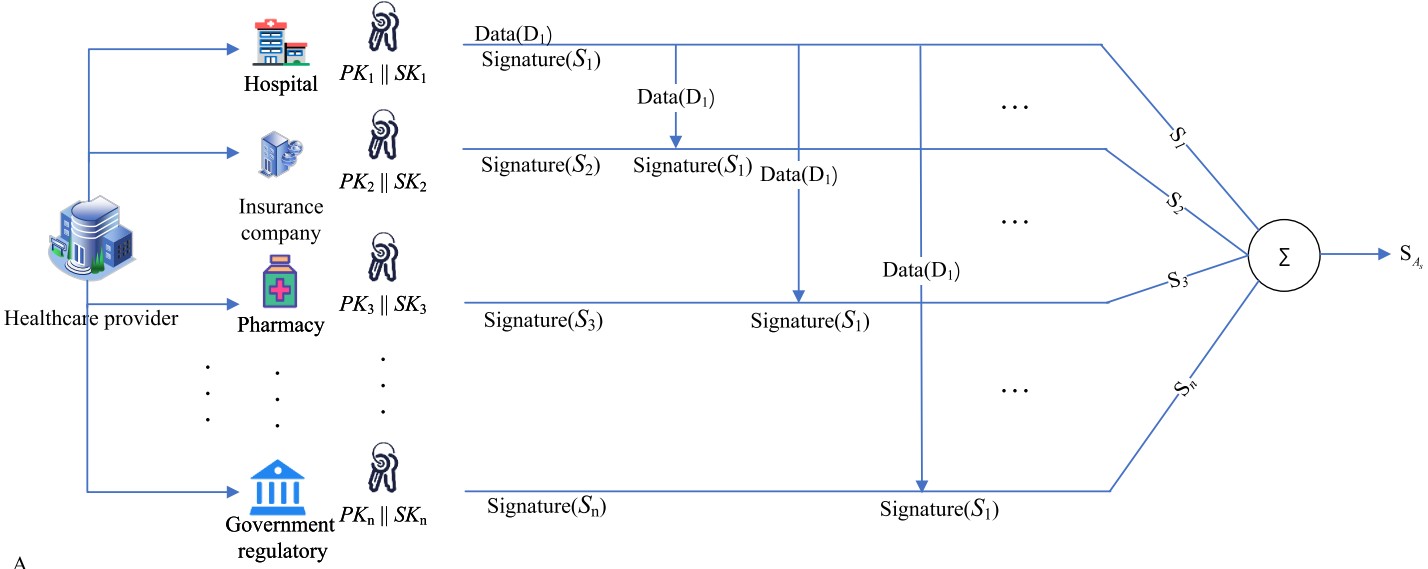

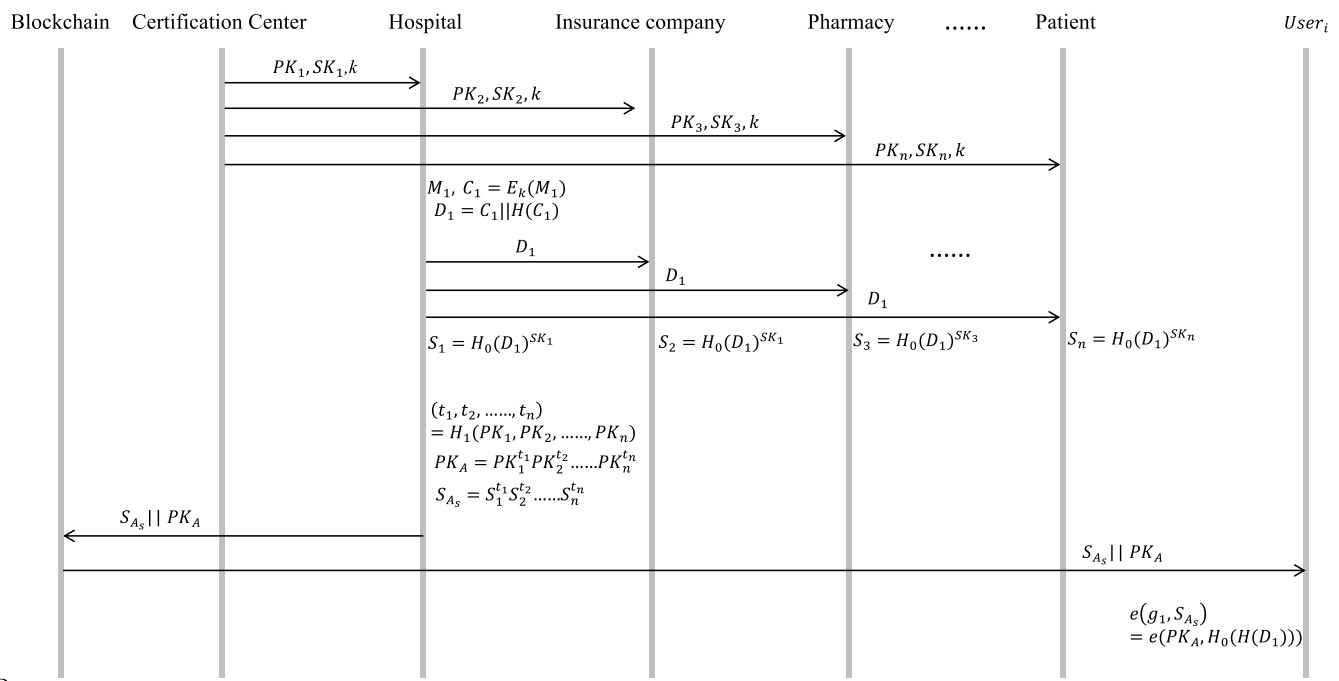

**Figure 4** (A) Cross-chain signature SAs. (B) Signature SAs process flow diagram. Healthcare provider, Hospital, Insurance company, Pharmacy, Government regulatory PK&SK: icons from Flaticon.com.

c) Sign $(D_1, SK_1)$

This algorithm takes the prepared data $D_1$ and the signing key $SK_1$ as inputs. Eventually, it returns the signature $S_1$ as a result.

$$S_1 \leftarrow H_0(D_1)^{SK_1} \in \mathbb{G}_0$$

d) Share data $D_1$ and sign

The patient shares the data $D_1$ with other healthcare providers, thus each provider has the same data $D_1$ and uses their signing key $SK_i$ to output different signatures $S_i$.

$$S_i \leftarrow H_0(D_1)^{SK_i} \in \mathbb{G}_0$$

e) Signature aggregate $((PK_1, S_1), (PK_2, S_2), \ldots, (PK_n, S_n))$

This algorithm takes all the individual signatures related to different users, then computes $t_1, t_2, \ldots, t_n$ and outputs the aggregation signature $S_{A_s}$.

$$(t_1, t_2, \ldots, t_n) \leftarrow H_1(PK_1, PK_2, \ldots, PK_n) \in R^n$$
$$S_{A_s} \leftarrow S_1^{t_1} \ldots S_n^{t_n} \in \mathbb{G}_0$$

f) Public key aggregate

This process involves advanced preparation for verifying the signature. The algorithm incorporates all the relevant individual public keys associated with different healthcare providers, then computes $t_1, t_2, \ldots, t_n$ and outputs the aggregation of the public key $PK_A$.

$$(t_1, t_2, \ldots, t_n) \leftarrow H_1(PK_1, PK_2, \ldots, PK_n) \in R^n$$
$$PK_A \leftarrow PK_1^{t_1}, PK_2^{t_2}, \ldots, PK_n^{t_n} \in \mathbb{G}_1$$

g) Verify $(H(D_1), PK_A, S_{A_s})$

This algorithm takes the hashed data $H(D_1)$, the aggregation of the public key $PK_A$, and the aggregation signature $S_A$ to verify if $e(g_1, S_A) = e(PK_A, H_0(H(D_1)))$ the output "accepts", or otherwise output "rejects".

The verification process is proven as below:

$$
\begin{aligned}
e(g_1, S_{A_s}) &= e(g_1, S_1^{t_1} \ldots S_n^{t_n}) = e(g_1, S_1^{t_1}) \cdot e(g_1, S_2^{t_2}) \cdot \ldots \cdot e(g_1, S_n^{t_n}) \\
&= e(g_1^{t_1}, S_1) \cdot e(g_1^{t_2}, S_2) \cdot \ldots \cdot e(g_1^{t_n}, S_n) \\
&= e(g_1^{t_1}, H_0(D_i)^{\alpha_1}) \cdot e(g_1^{t_2}, H_0(D_i)^{\alpha_2}) \cdot \ldots \cdot e(g_1^{t_n}, H_0(D_i)^{\alpha_n}) \\
&= e((g_1^{t_1})^{\alpha_1}, H_0(D_i)) \cdot e((g_1^{t_1})^{\alpha_2}, H_0(D_i)) \cdot \ldots \cdot e((g_1^{t_1})^{\alpha_n}, H_0(D_i)) \\
&= e((g_1^{\alpha_1})^{t_1}, H_0(D_i)) \cdot e((g_1^{\alpha_2})^{t_1}, H_0(D_i)) \cdot \ldots \cdot e((g_1^{\alpha_n})^{t_1}, H_0(D_i)) \\
&= e(PK_1^{t_1}, H_0(D_i)) \cdot e(PK_2^{t_2}, H_0(D_i)) \cdot \ldots \cdot e(PK_n^{t_n}, H_0(D_i)) \\
&= e(PK_1^{t_1} \ldots PK_n^{t_n}, H_0(D_i)) \\
&= e(PK_A, H_0(D_i))
\end{aligned}
$$

***Aggregate signature $S_{A_f}$***

The process of obtaining the signature $S_{A_f}$ is illustrated in Fig. 5A, and the protocol flow is shown in Fig. 5B. This process is generally similar to the one described above, with the signing data being the privacy data generated by User 1 at different times. At time $T_n$, all signatures of User 1 are $S_{1T_1}, S_{1T_2}, \ldots, S_{1T_n}$. The aggregate signature $S_{A_{f_1} T_n}$ is then computed as $S_{A_{f_1} T_n} \leftarrow S_{1T_1}{}^{t_1} \ldots \ldots S_{1T_1}{}^{t_n}$.

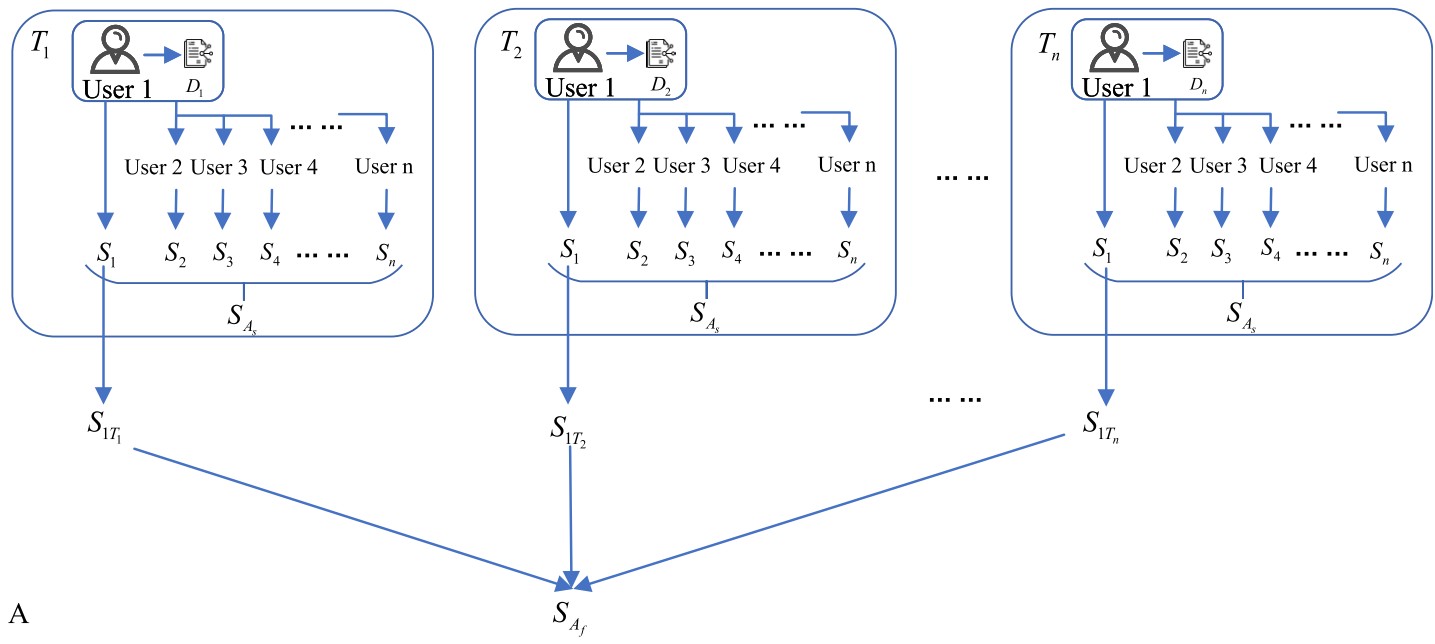

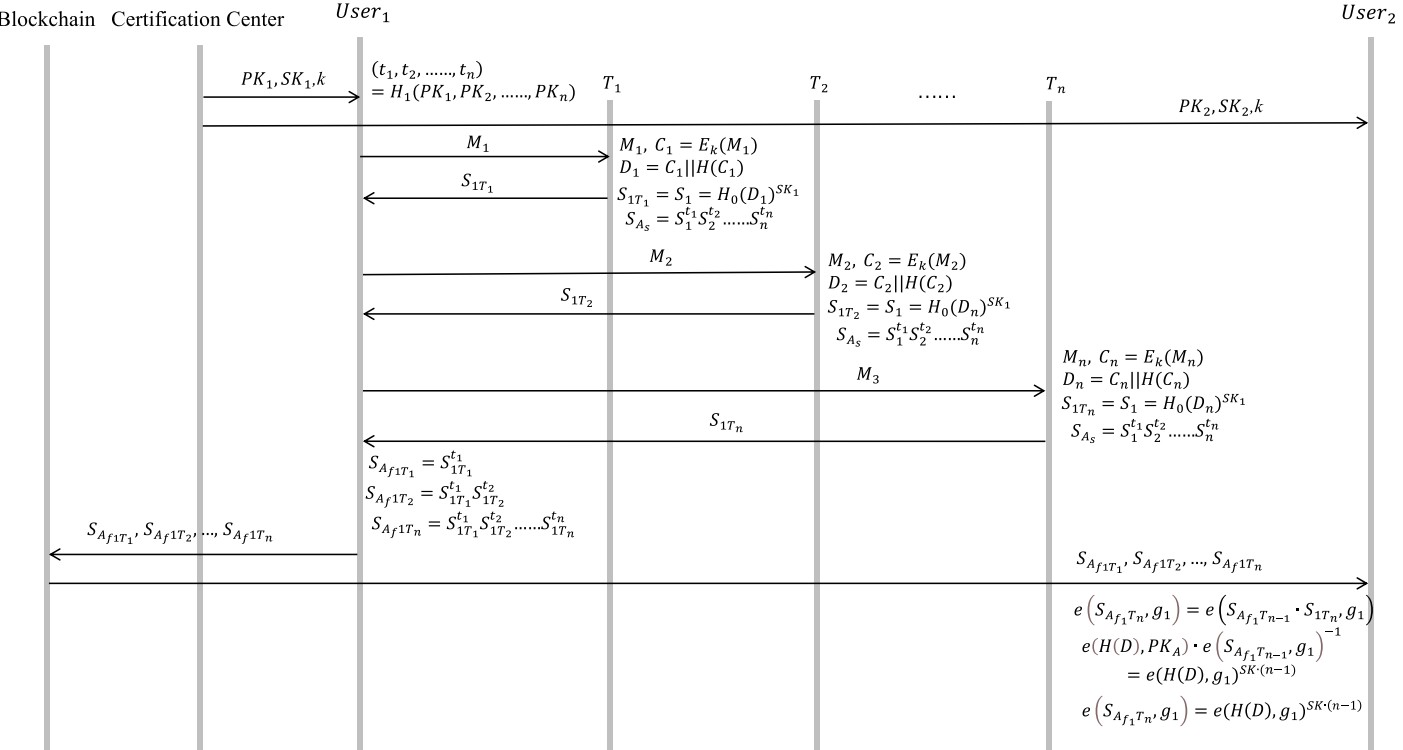

**Figure 5 (A) Cross-chain signature SAf. (B) Signature SAf process flow diagram.** User, Di: icons from flaticon.com.

The property of aggregate signatures, which allows for the verification of individual signature existence, is utilized to create an invisible chain formed by the aggregate signatures. This enables the identification and categorization of all data uploaded by a particular user from the mixed data, achieving the fitting of a specific type of data. This not only organizes the system's data more effectively but also enhances the efficiency and accuracy of data management.

Suppose User 1 updates data at time $T_n$ and generates the aggregate signature $S_{A_{f_1} T_n}$. This data and signature are then uploaded to the off-chain IPFS distributed storage system. Additionally, at times $T_{n-1}, T_{n-2}, \ldots, T_1$, the system stores aggregate signatures of a large amount of user data, denoted as $S_{A_{f_i} T_j}$. All these signature sets are referred to as $\mathcal{H}$. For $\forall S_{A_{f_i} T_j} \in \mathcal{H}$, the system can search and verify all data generated by User 1 at times $T_j < T_n$ within the aggregate signatures. The existence of $S_{A_{f_1} T_j}$ can be confirmed through effective aggregation. The verification formula is $e\left(S_{A_{f_1} T_n}, g_1\right) = e(H(D), PK_A) \cdot e\left(S_{A_{f_1} T_{n-1}}, g_1\right)^{-1}$.

a) The aggregate signature $S_{A_{f_1} T_n}$ and the aggregate public key $S_{A_{f_1} T_n}$ are considered. Suppose $S_{A_{f_1} T_{n-1}}$ is a part of the aggregate signature $S_{A_{f_1} T_n}$.

$$
e\left(S_{A_{f_1} T_n}, g_1\right)
$$
$$
= e(S_{1T_1} \cdot S_{1T_2} \cdot \ldots \ldots \cdot S_{1T_n}, g_1)
$$
$$
= e\left(S_{A_{f_1} T_{n-1}} \cdot S_{1T_n}, g_1\right)
$$

b) Next, the following computation is performed:

$$
e(H(D), PK_A) \cdot e\left(S_{A_{f_1} T_{n-1}}, g_1\right)^{-1}
$$
$$
= e\left(H(D), PK_1^n\right) \cdot e\left(S_{A_{f_1} T_{n-1}}, g_1\right)^{-1}
$$
$$
= e(H(D), PK_1 \cdot PK_1 \cdot \ldots \ldots \cdot PK_1) \cdot e\left(S_{A_{f_1} T_{n-1}}, g_1\right)^{-1}
$$
$$
= e(H(D), g_1)^{SK \cdot n} \cdot e(SK \cdot H(D), g_1)^{-1}
$$
$$
= e(H(D), g_1)^{SK \cdot n} \cdot e(SK \cdot H(D), g_1)^{-SK}
$$
$$
= e(H(D), g_1)^{SK \cdot (n-1)}
$$

Here, $e\left(S_{A_{f_1} T_{n-1}}, g_1\right)^{-1}$ is used to "cancel out" the contribution of $S_{A_{f_1} T_{n-1}}$ in the aggregate signature. If the equation holds, then it can be concluded that the signature $S_{A_{f_1} T_{n-1}}$ is indeed a part of the aggregate signature $S_{A_{f_1} T_n}$.

c) Finally, the expression $e\left(S_{A_{f_1} T_n}, g_1\right) = e(H(D), g_1)^{SK \cdot (n-1)}$ is obtained, which is equal to the left-hand side of the equation, indicating that the signature $S_{A_{f_1} T_{n-1}}$ is a part of the aggregate signature $S_{A_{f_1} T_n}$.

## System operation details

### Patient blockchain: personal information addition

When a patient is initially registered in the system, they are required to provide basic personal information. To ensure privacy, the CP-ABE encryption method is utilized to

**Peer**J Computer Science

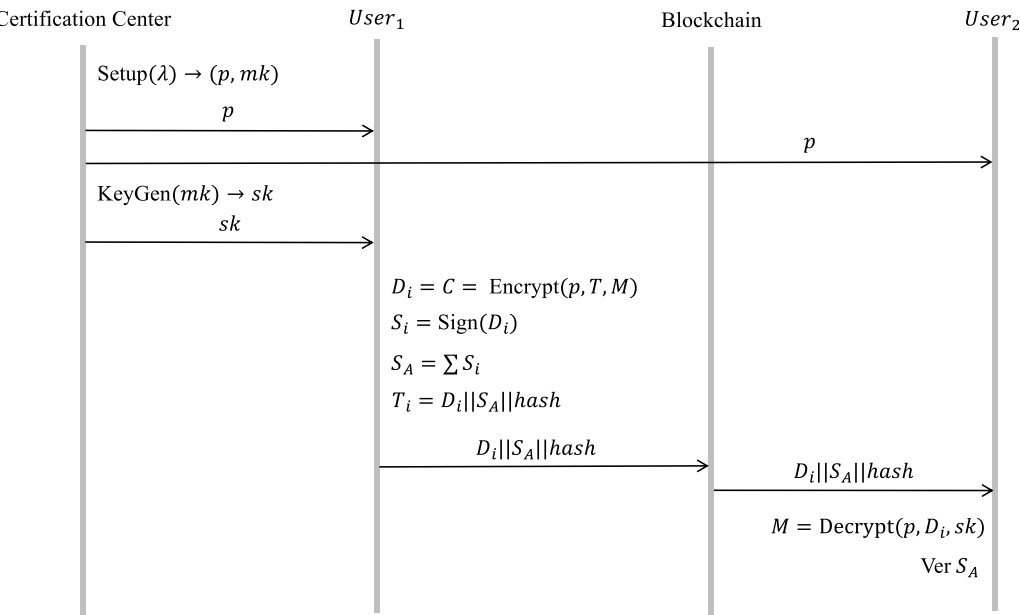

**Figure 6 Personal information submission and encryption protocol flow.**

encrypt the patient's private data. The encrypted data is uploaded into IPFS, where a hash value is generated and then transferred to the patient blockchain along with the signature $S_A$. The corresponding process flow is illustrated in Fig. 6.

a) Setup $(\lambda) \rightarrow (pk, mk)$

This procedure takes the security parameter $\lambda$ as input and produces the public parameter $p$ and master key $mk$ for the proposed CP-ABE mechanism.

b) KeyGen $(mk, A) \rightarrow sk$

This procedure takes the master key $mk$ and the attribute set $A$ as input and generates the user attribute secret key $sk$.

c) Encrypt $(p, T, M) \rightarrow C$

This procedure takes the public parameter $p$, accesses the structure $T$ and the patient's personal information $M$, and encrypts the plaintext $M$ into the ciphertext $C$.

d) Sign $(C) \rightarrow S_A$

In this process, the user generates shareable data $D_i$ using the signature algorithm described above and signs $D_i$ to obtain the corresponding signed data $S_A$.

e) Data upload to the blockchain

The signature $S_A$ and the *hash* value returned by IPFS are incorporated into a data transaction $T_i = D_i||S_A||hash$ and subsequently uploaded to the patient blockchain.

f) Decrypt $(p, C, sk) \rightarrow M$

This procedure takes the public parameter $p$, ciphertext $C$, and secret key $sk$. as input and generates the plaintext $M$.

### *Healthcare provider blockchain: health record addition*

Healthcare data is decentralized among a variety of healthcare providers, and encompasses entities such as hospitals, insurers, pharmacies, and governmental regulatory bodies.

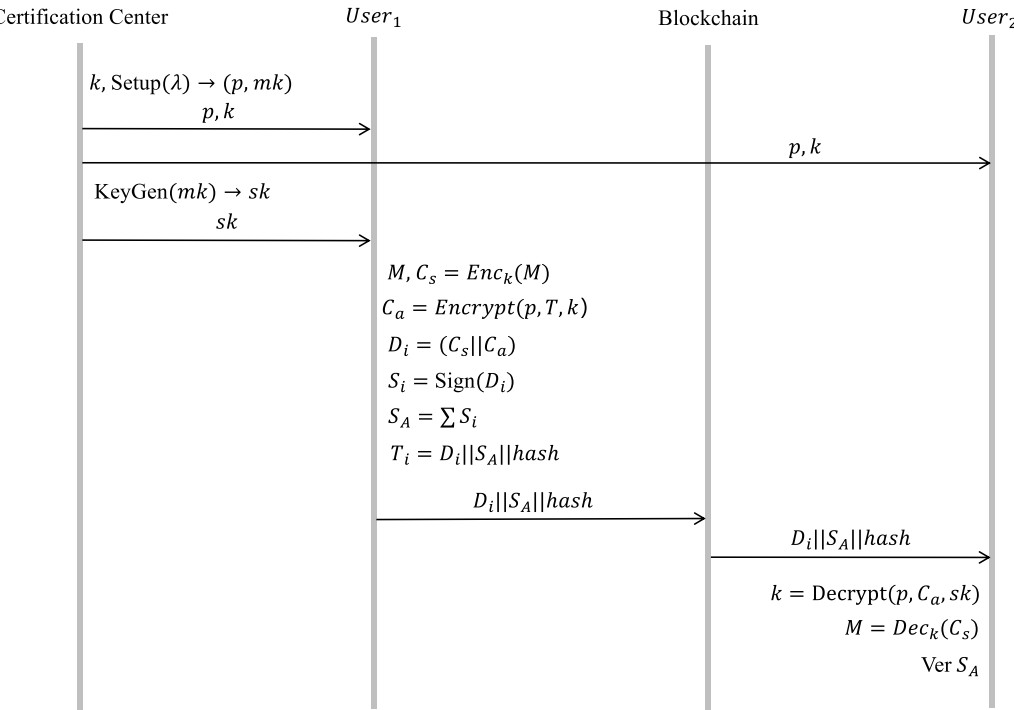

**Figure 7 Health record submission and encryption protocol flow.**

Different from the ciphertext present in the patient blockchain, this medical data is considerably more substantial in volume. Initially, the medical data is encrypted using symmetric encryption. Subsequently, the symmetric encryption key itself is encrypted *via* CP-ABE. The final encrypted data can be procured by concatenating these two ciphertexts. The process is illustrated in Fig. 7.

a) Key generation

When a user affiliated with a healthcare provider partakes in the system, a symmetric encryption key $k$ is allocated by the system. The key assignment for CP-ABE is the same as that in the patient blockchain and is not described in detail here.

b) Encrypt $(k, p, T, M) \rightarrow C$

Within this process, the healthcare provider generates the medical data $M$, which is encrypted using the key $k$, resulting in $C_s = Enc_k(M)$. Following this, attribute encryption is performed on the key $k$ $Encrypt(P, T, k) \rightarrow C_a$. $(C_s || C_a)$ which is the final ciphertext data $C$.

c) Sign $(C) \rightarrow S_A$

In this process, the user generates shareable data $D_i$ using the signature algorithm described above and signs $D_i$ to obtain the corresponding signed data $S_A$.

d) Data upload to the blockchain

The signature $S_A$ and the *hash* value returned by IPFS are incorporated into a data transaction $T_i = D_i || S_A || hash$ and subsequently uploaded to the healthcare provider blockchain.

e) Decrypt $(p, C, sk, k) \rightarrow M$

This procedure takes the public parameter $p$, the ciphertext $C_a$, and the secret key $sk$. as input and generates the symmetric encryption key $k$. The key $k$ is subsequently employed to decrypt the ciphertext $C_s$, facilitating the retrieval of the original data $M$.

### Social blockchain: data sharing externally

The social blockchain establishes a connection with the external blockchain system to facilitate the sharing of medical data among different healthcare institutions. To ensure proper data sharing and protection, a data processor categorizes the medical data into five levels of sensitivity. This approach aligns closely with the principles of ISO/IEC 27001 Information Security Management (*ISO/IEC, 2022*), which emphasizes the classification of information based on its importance, sensitivity, and associated risks. By implementing this structure, the classification ensures that appropriate protection measures are applied to safeguard the data's confidentiality, integrity, and availability.

The five-level classification reflects a progressive sensitivity model, gradually increasing the restrictions and controls as the data moves from being fully public to highly sensitive. This clear and logical progression aligns with ISO/IEC 27001's principle of risk-based classification and access control. Each level is designed to mitigate risks associated with unauthorized access or misuse, ensuring that data is accessed only by authorized entities and under defined conditions.

Here is a detailed explanation of the five levels of sensitivity:

Level 1: Fully public data. This category includes information such as the hospital's name, address, and telephone number, which can be openly shared with the public on the Internet without any restrictions or privacy concerns.

Level 2: Data available for widespread access. This category is comprised of data that can be accessed on a large scale, typically after obtaining approval through a formal application process. These datasets are often made available for research and analysis purposes, facilitating scientific investigations and advancements in various domains.

Level 3: Data available for restricted access. This category includes data that can be accessed on a medium scale, typically limited to usage within the authorized project team operating under the purview of a specific institution. Access to this data is granted only to team members involved in the project, ensuring compliance with institutional policies and safeguarding the privacy and security of the data.

Level 4: Data available for limited access. This category pertains to data that can be accessed on a smaller scale, specifically restricted to individuals directly involved in the consultation process. Access to this data is confined to healthcare professionals and relevant stakeholders who require access to providing healthcare services and facilitating the consultation. Strict confidentiality measures are implemented to protect the privacy and sensitivity of the data.

Level 5: Data available for highly restricted access. This category encompasses data that can only be accessed on a very limited scale and under stringent restrictions. For instance, specific disease-related information, such as on Acquired Immune Deficiency Syndrome (AIDS) or sexually transmitted diseases (STDs), is strictly limited to access by primary care

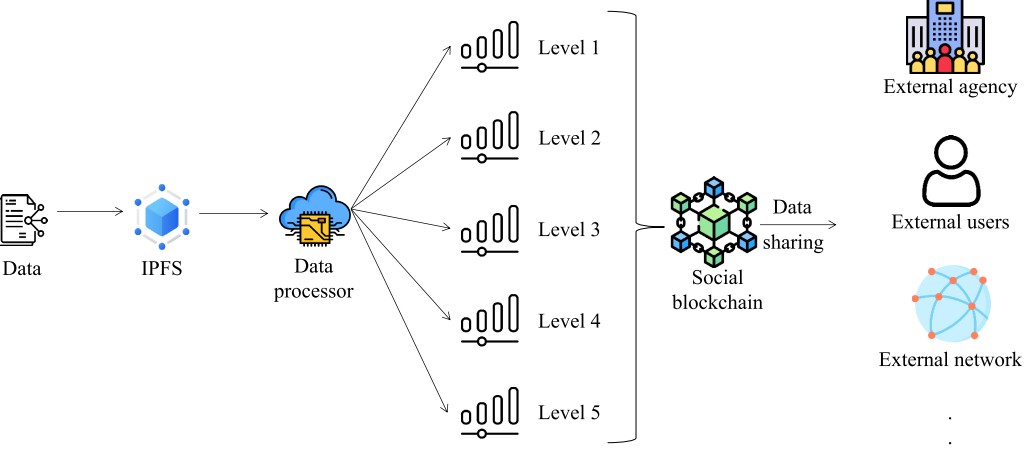

**Figure 8  Data-sharing process.** Data, IPFS, Data Processor, Social blockchain, Level, External agency, External users, External network: icons from flaticon.com.

---

**Algorithm 1:**  **Medical data (MD) categorization based on sensitivity level.**

**Input:** MD

**Output:** Categorized data (CD) with sensitivity level (SL)

\# Stage I: Determining Sensitivity Level

1: Determine the SL of RMD based on predefined criteria → SL

\# Stage II: Categorizing data

2: CD = { 'data': MD, 'sensitivity': SL }

\# Stage III: Returning categorized data

3: Return CD

---

providers who require the data for clinical purposes. Comprehensive controls and protocols are in place to ensure the utmost confidentiality and privacy protection of this sensitive information, adhering to regulatory guidelines and ethical considerations.

The data-sharing process is shown in Fig. 8. When an external user requires access to medical data, the system employs an automated process to determine the appropriate levels of data accessibility based on the user's attributes. Upon identification, the user can retrieve the corresponding hash from the social blockchain, which serves as a reference for obtaining the authorized data from the IPFS storage system. This dynamic approach ensures that users can securely retrieve and access the specific data they are permitted to view, maintaining the privacy and confidentiality of the overall healthcare ecosystem.

Two algorithms have been conceptualized, specifically tailored for the tasks of automatic data hierarchy creation and data dissemination. Algorithm 1 illustrates the mechanism of data categorization. This mechanism involves a detailed stratification of data according to sensitivity levels, thereby differentiating information that is permissible for open distribution from datasets that demand heightened sensitivity.

| Algorithm 2: Medical data sharing *via* social blockchain. |
| --- |
| **Input:** Categorized Data CD, User User |
| **Output:** Hash value (HV) or none |
| # Stage I: Checking user permission |
| 1: Check user permission for CD ['sensitivity'] → Permission |
| 2: If permission is false → Return none |
| # Stage II: Retrieving HV from IPFS and uploading to social blockchain |
| 3: Retrieve HV from IPFS for CD ['data'] → HV |
| 4: Upload HV to social blockchain → Record |
| 5: Return HV |

Algorithm 2 elaborates the process of medical information distribution leveraging social blockchain technology. This process necessitates an evaluation of user permissions, only those users satisfying the specified criteria are granted data access. Furthermore, the data distribution procedure is inscribed in the framework of the social blockchain infrastructure, thereby establishing unequivocal transparency and traceability.

# EXPERIMENTS AND ANALYSIS
## Safety analysis
### Provable security

**Definition:** In a prime-order cyclic group $\mathbb{G}$, given a generator $g$ and elements $g^a, g^b$, where $a, b \in Z_q^*$, the task of computing $g^{ab}$ considered computationally infeasible.

Our algorithm utilizes bilinear pairings and aggregate signatures to enhance data security. By analyzing the difficulty of forging signatures under the CDH assumption, we demonstrate the security of the scheme.

a. Threat model

An attacker $\mathcal{A}$ attempts to forge a valid aggregate signature $S_A$ without knowledge of the private key $SK_i$, such that the verifier accepts the equation:

$$e(g_1, S_A) = e(PK_A, H_0(H(D)))$$

where $g_1$ is the generator of group $\mathbb{G}_1$, $S_A$ is the aggregate signature, $PK_A$ is the aggregate public key, $H_0$ is the hash function mapping data to group $\mathbb{G}_0$, and $H(D)$ is the hash value of the data $D$.

b. Difficulty of signature forgery based on the CDH assumption

**Lemma 1:** Under the CDH assumption, the probability of an attacker forging a valid aggregate signature $S_A$ without knowledge of the private key is negligible.

Suppose there exists an attacker $\mathcal{A}$ that can forge a valid aggregate signature $S_A$ with non-negligible probability $\varepsilon_{forge}$. We construct a simulator $\mathcal{B}$ that uses $\mathcal{A}$ to solve the CDH problem.

Simulator $\mathcal{B}$:

- Input: A CDH problem instance $(g, g^a, g^b)$, The goal is to compute $g^{ab}$.
- Setup:

  Set $g_1 = g$, public keys $PK_1 = g^a$, and $PK_2 = g^b$. The simulator does not know the corresponding private keys $SK_1 = a$ and $SK_2 = b$.

- Challenge:

  Choose a random message $D$, compute its hash value $H(D)$, and compute $H_0(H(D))$.

- Interaction with the Attacker:

  Provide $\mathcal{A}$ with $PK_1, PK_2$, and $H_0(H(D))$, and request $\mathcal{A}$ to output a forged aggregate signature $S_A$.

- Attacker's Output:

  If $\mathcal{A}$ outputs a signature $S_A$ such that:

$e(g_1, S_A) = e(PK_A, H_0(H(D)))$

where $PK_A = PK_1{}^{t_1} PK_2{}^{t_2}$ and $t_1, t_2$ are known coefficients.

- Calculation:

  Based on the verification equation and the properties of bilinear pairings, we compute:
  $e(g_1, S_A) = e(PK_1^{t_1} PK_2^{t_2}, H_0(H(D))) = e(PK_1, H_0(H(D)))^{t_1} \cdot e(PK_2, H_0(H(D)))^{t_2}$
  Substituting $PK_1 = g^a, PK_2 = g^b$, we get:

$e(g_1, S_A) = e(g, H_0(H(D)))^{at_1 + bt_2}$

- Extracting $g^{ab}$:

  Simulator $\mathcal{B}$ cannot directly extract $g^{ab}$ from $at_1 + bt_2$. However, if $\mathcal{A}$ is able to forge a signature with non-negligible probability, it implies that $\mathcal{B}$ can solve the CDH problem with the same probability, which contradicts the assumption.

  Since, under the CDH assumption, the probability of solving the above problem is negligible, the probability of the attacker successfully forging a signature is also negligible.

$\varepsilon_{forge} \leq \varepsilon_{CDH}$

where $\varepsilon_{CDH}$ is the negligible probability of solving the CDH problem.

### Blockchain security

Attacks on blockchain systems are typically centered around the concept of computational power. Specifically, in a 51% attack, when an attacker controls more than 50% of the total computational power in the blockchain network, they can potentially compromise the system by rewriting the blockchain, double-spending coins, or invalidating transactions.

This article analyzes the security of blockchain systems based on the gambling probability attack model.

In this article, we assume that a malicious attack node (MAN) successfully joins the proposed blockchain-based EMR system and launches an attack on any of the chains, causing a consensus node on that chain to fork. The health node (HN), which is responsible for maintaining the integrity of the data and ensuring the security of the blockchain, is also involved in the consensus process. According to the Bitcoin whitepaper, blockchain nodes always follow the longest chain as the correct one and extend it. Therefore, if the MAN can control more than 50% of the computational power, it can generate a longer chain to replace the honest nodes' chain, thereby succeeding in the attack. This attack process can be viewed as a binomial random walk. The specific discussion of this process is as follows:

$$
\begin{cases}
1 & , k > z \\
\left(\dfrac{q}{p}\right)^{z-k} & , k \leq z
\end{cases}
$$

where $k$ represents the number of blocks successfully mined by MAN, $z$ denotes the number of blocks mined by the honest nodes before MAN attempts to replace the chain, $q$ s the probability of MAN successfully mining a block (representing the attacker's computing power proportion), and $p$ is the probability of the honest nodes mining a block (representing the honest nodes' computing power proportion, where $p + q = 1$).

As $k$ can be any non-negative integer, the probability distribution of $k$ follows a Poisson distribution, which is calculated as follows:

$$
\sum_{k=0}^{\infty} \frac{\lambda^k e^{-\lambda}}{k!}
$$

where $\lambda$ represents the expected number of events (blocks mined) occurring in the given time interval. It is the mean of the distribution.

Therefore, the formula for calculating the probability of MAN successfully attacking the chain is as follows:

$$
P = \sum_{k=0}^{\infty} \frac{\lambda^k e^{-\lambda}}{k!} \cdot
\begin{cases}
1 & , k > z \\
\left(\dfrac{q}{p}\right)^{z-k} & , k \leq z
\end{cases}
$$

The formula after simplification is:

MAN uses a "gambling attack" to attack the EMR system in order to obtain all EMR data. The probability of a successful attack depends mainly on the probability of MAN generating the next block and the number of "chains" used in the blockchain system. This solution combines smart healthcare with a three-chain structure, while the solutions presented in *Shuaib et al. (2022)* and *Guo et al. (2022)* are single-chain structures, and the solutions presented in *Liu et al. (2020, 2022)* are both double-chain structures.
Based on the above analysis, the specific experimental plan of this article is to have MAN generate the next block with probabilities of 0.10, 0.15, 0.20, 0.25, 0.30, 0.35, 0.40, and 0.45 respectively. These probabilities were selected to model a range of realistic scenarios, from low-intensity attacks where MAN has minimal computing power $q = 10$ to high-intensity attacks nearing the critical threshold $q = 0.45$. This range reflects the typical distribution of computational power in blockchain networks, where attackers rarely possess more than half of the network's total power. The incremental increase of 0.05 between values ensures sufficient granularity to capture the system's response under varying levels of attack intensity, enabling a comprehensive evaluation of the proposed system's resilience.

The selection of probabilities from 0.10 to 0.45 was based on the following considerations:

a. Representativeness: These probabilities cover a range from low to high attack success rates, fully reflecting the system security under different levels of attack intensity.

b. Practical application: In actual blockchain applications, attackers usually find it challenging to obtain extremely high computing power. Therefore, selecting probabilities that increment from low values is more in line with real-world scenarios.

c. Scientific basis: The commonly used blockchain attack models in the literature also adopt similar probability ranges, ensuring the comparability and scientific validity of our experiments.

The minimum secure block number that HN needs to generate is then calculated and counted for the EMR system of the blockchain to ensure the attack success probability is below 0.1, 0.01, and 0.001 respectively.

In the experimental design, several potential confounding variables were controlled to ensure the reliability of the results:

a. Node type and number: The type and number of nodes used in all experiments were kept consistent.

b. System load: The system load was controlled throughout the experiments to maintain uniform starting conditions.

c. Environmental settings: All experiments were conducted in the same hardware and software environment to avoid discrepancies due to equipment differences.

d. Repetition of experiments: Each set of experiments was repeated multiple times, and the average value was taken to reduce random errors.

To implement the theoretical framework described earlier, we used MATLAB to calculate the attack success probabilities and the corresponding minimum number of secure blocks required. Specifically, we applied the Poisson distribution and the formulas outlined earlier to model the attack scenarios. Using the thresholds for attack success probabilities of 0.1, 0.01, and 0.001, the MATLAB code calculates the minimum number of secure blocks that HN need to generate to ensure the attack success probability remains below each of these thresholds.

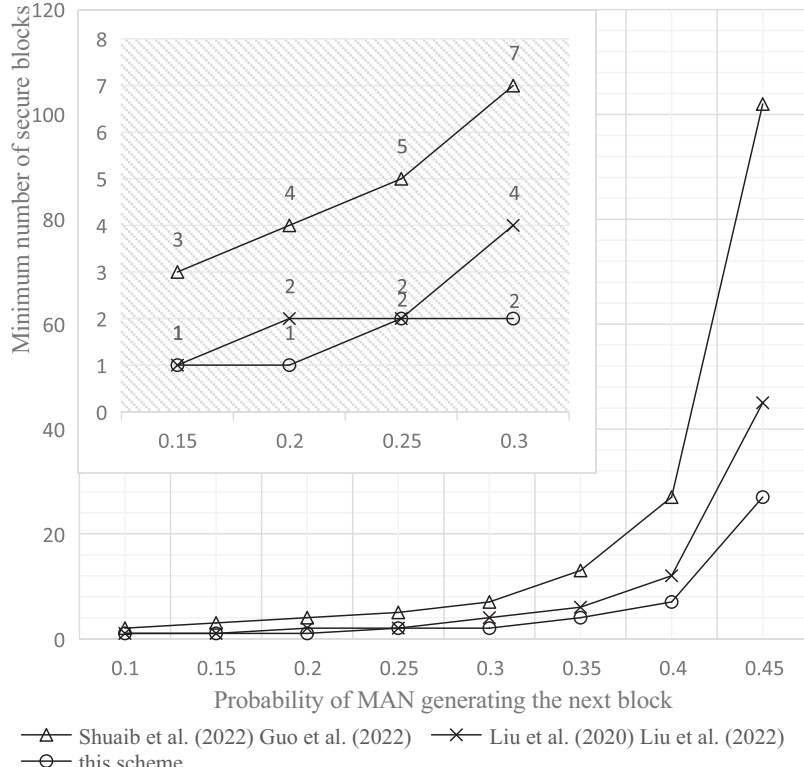

**Figure 9 Comparison of the minimum number of safe blocks.** Studies: *Shuaib et al. (2022)*, *Guo et al. (2022)*, *Liu et al. (2020, 2022)*; this study.

The experimental tools and simulation environment used in this article were a personal computer with an Intel(R) Core(TM) i5-1035G1 CPU @ 1.00 GHz 1.19 GHz processor, 8.00 GB memory, and a 64-bit Windows operating system. The experimental results are shown in Figs. 9–11.

a) Probability of MAN attack below 0.1

Experimental results presented in Fig. 9 demonstrate that to maintain the probability (P) of a successful MAN attack on the blockchain system below 0.1; this scheme's increase in the minimum number of secure blocks required is smaller than that of other comparison schemes once the MAN's block generation probability reaches 0.2. Furthermore, elevating the MAN's block generation probability to 0.45 leads to a substantial enhancement in security. Specifically, the required minimum number of secure blocks is lowered by 73.5% in comparison to the single-chain scheme, and by 44.4% relative to the double-chain scheme.

b) Probability of MAN attack below 0.01

Experimental results depicted in Fig. 10 indicate that, to maintain the probability (P) of a successful MAN attack on the blockchain system below 0.01, increasing the MAN's block generation probability to 0.45 significantly enhances security. Specifically, this adjustment

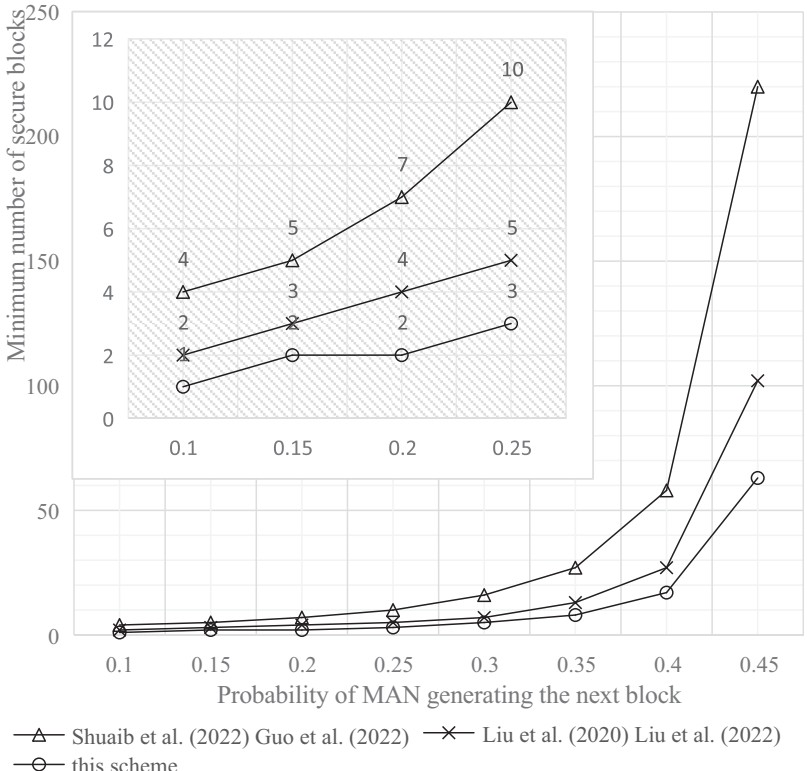

**Figure 10 Comparison of the minimum number of safe blocks at P0.01.** Studies: *Shuaib et al. (2022)*, *Guo et al. (2022)*, *Liu et al. (2020, 2022)*; this study.

reduces the minimum number of secure blocks required by 71.3% compared to the single-chain scheme and by 38.2% when compared to the double-chain scheme.

c) Probability of MAN attack below 0.001

Experimental findings presented in Fig. 11 reveal that, to keep the probability (P) of a MAN successfully attacking the blockchain system under 0.001, the increase in the minimum number of secure blocks required at the genesis block by this scheme is less than that required by comparison schemes. Further, when the probability of MAN generating the next block is raised to 0.45, there is a significant improvement in security. Specifically, the minimum number of secure blocks needed by this scheme is reduced by 70.0% compared to the single-chain scheme, and by 36.2% in comparison to the double-chain scheme.

The experimental results clearly demonstrate that our approach significantly enhances the security of blockchain systems. By reducing the minimum number of secure blocks required to keep the attack success probability below the thresholds of 0.1, 0.01, and 0.001, our method reduces the required number of blocks by approximately three times compared to existing schemes, thereby significantly improving the system's resilience against 51% attacks.

**Peer**J Computer Science

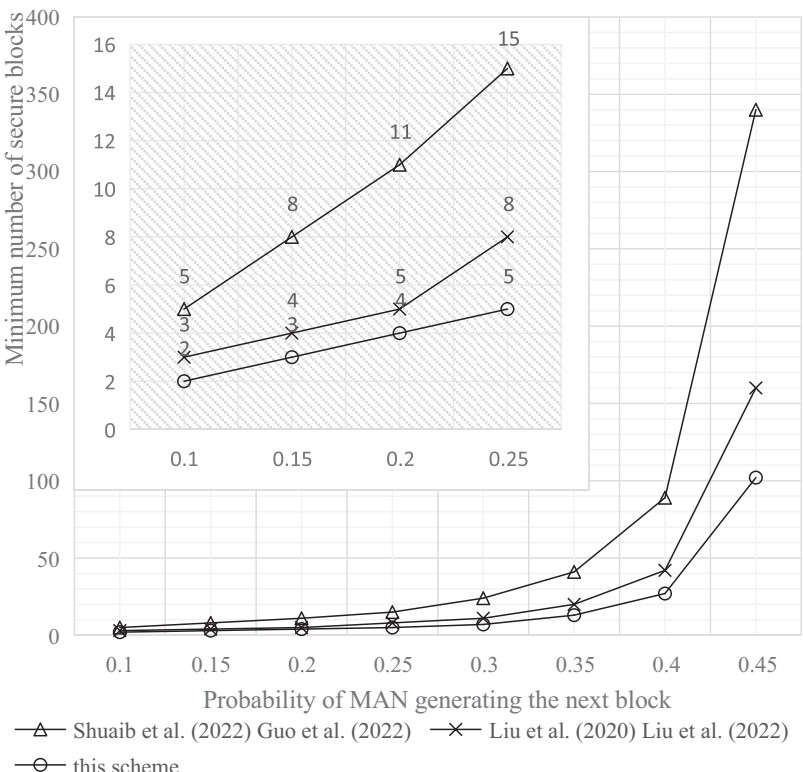

**Figure 11 Comparison of the minimum number of safe blocks at P0.001.** Studies: *Shuaib et al. (2022)*, *Guo et al. (2022)*, *Liu et al. (2020, 2022)*; this study.

## System performance analysis

### Performance evaluation

To further valaaidate the scheme, the Faker library was used to generate random medical data, and smart contracts were deployed on a local Ethereum network using Solidity. The evaluation focused on transaction latency and throughput, as these metrics are widely recognized as critical indicators of a blockchain system's responsiveness, operational efficiency, and scalability. Furthermore, the emphasis on these metrics aligns with the standard practices in blockchain performance research, where transaction-related benchmarks are considered foundational for assessing system performance.

A sample size of 1,000 transactions was selected to evaluate the performance of the system. This choice was based on the following considerations:

a) Representativeness: A sample size of 1,000 transactions is large enough to capture the variability in transaction latency and throughput under typical operating conditions, providing a comprehensive assessment of the system's performance.

b) Practical application: In real-world blockchain applications, it is common to handle a large number of transactions. Testing with 1,000 transactions ensures that the evaluation reflects realistic usage scenarios and can provide insights into how the system performs under substantial load.

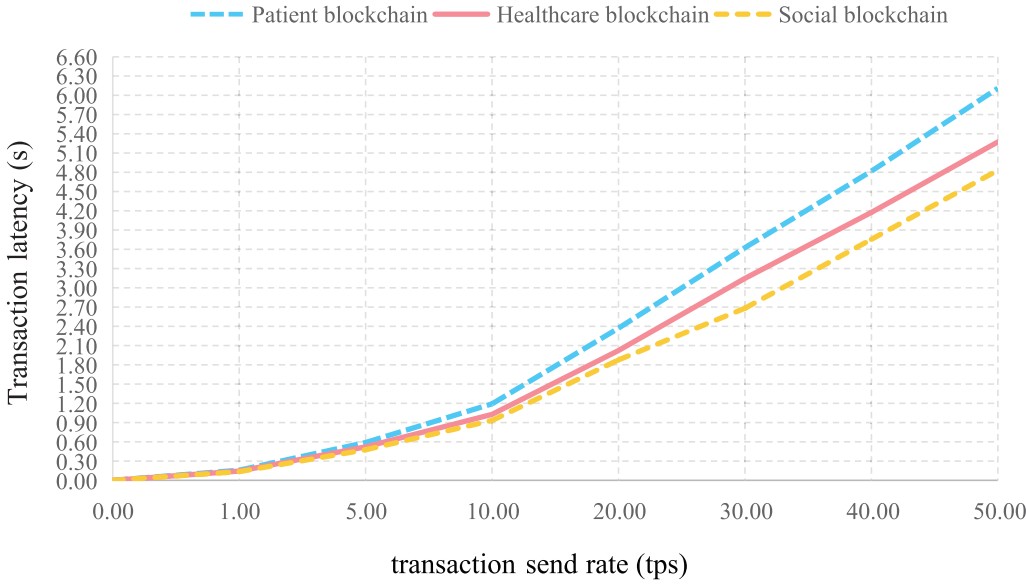

**Figure 12 Latency *vs.* number of transactions for each blockchain.**

c) Scientific basis: Prior studies and benchmarks in blockchain performance testing often utilize similar or smaller sample sizes to evaluate system performance metrics, ensuring the comparability and scientific validity of our experiments.

By varying transaction send rates, the average latency per transaction was tested on the Patient blockchain, Healthcare provider blockchain, and Social blockchain. As shown in Fig. 12, latency increases with the number of transactions, remaining under 6 s at 50 transactions per second (tps). The Social blockchain exhibits slightly lower latency compared to the Patient and Healthcare provider blockchains, attributed to the pre-classification of data, allowing only low-sensitivity data to be shared.

Furthermore, the maximum and minimum throughput of the Patient blockchain, Healthcare provider blockchain, Social blockchain, and the entire system were evaluated at transaction send rates ranging from 10 to 40 tps. As shown in Fig. 13, the throughput across different transaction send rates concentrates around 35 tps.

In the performance testing, several potential confounding variables were controlled to ensure the validity of the results:
a) Data generation: The Faker library was used to generate random but consistent medical data across different experiments.
b) Smart contract deployment: All smart contracts were deployed using the same configuration and version of Solidity on the same local Ethereum network to maintain consistency.
c) System environment: The tests were conducted on the same hardware and software environment used in the block security comparison tests, ensuring uniformity.
d) Repetition and averaging: Each performance test was repeated multiple times, and the results were averaged to minimize random variations.

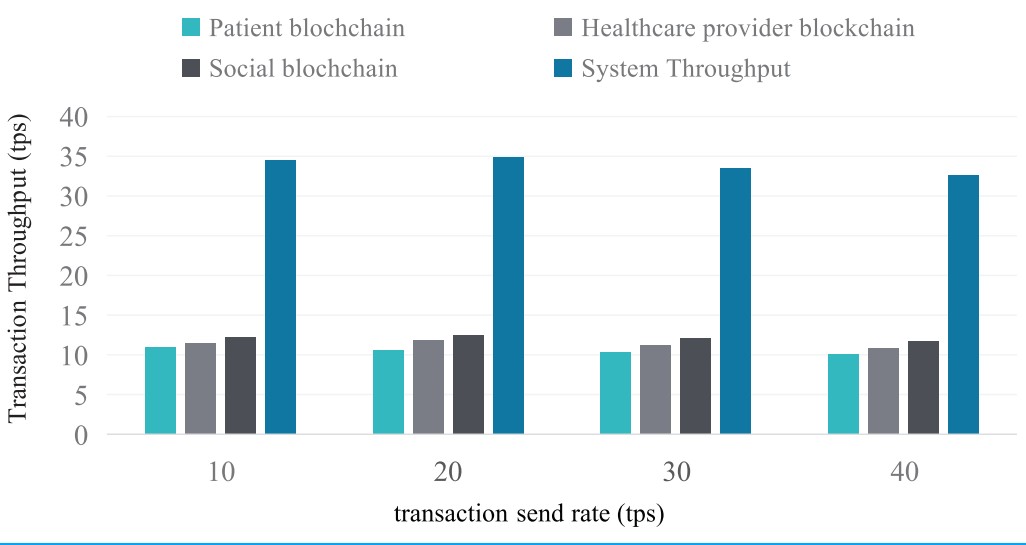

**Figure 13 Throughput performance.**   

Our findings indicate that while the use of blockchain introduces some overhead, it does not significantly degrade the overall system performance. Specifically, the latency and throughput metrics remain within acceptable ranges, demonstrating that the integration of blockchain is feasible without compromising efficiency. More importantly, the blockchain-based system significantly enhances data security and integrity, offering robust protection against tampering and unauthorized access.

These results underscore the practicality and feasibility of using blockchain in our system. The enhanced security and data integrity provided by blockchain outweigh the modest increase in complexity and resource consumption, making it a valuable addition to our EMR system.

### Storage optimization analysis

In addition to the performance metrics, storage optimization is another key benefit of the proposed system. The use of aggregated signatures has led to significant storage savings. Both types of signatures essentially combine multiple individual signatures into a single one, compressing storage requirements from linear growth to a constant size.

Aggregated signatures compress multiple signatures into a single one using bilinear pairings and hash functions. Without aggregation, the system would require storing individual signatures, consuming significantly more storage space. With aggregated signatures, only one signature is needed, reducing the storage requirement substantially.

Assuming each signature is 25 bytes, the storage requirements and savings for different numbers of signatures are shown in Table 3.

As the number of signatures increases, the storage savings rapidly approach 100%, making this approach particularly suitable for scenarios requiring the storage of a large number of signatures.

Moreover, aggregated signatures not only optimize storage but also significantly improve the efficiency of signature verification. In traditional methods, each signature

**Table 3  Storage requirements and savings with aggregated signatures.**

| Number of signatures | Non-aggregated storage requirement (bytes) | Aggregated storage requirement (bytes) | Storage savings (bytes) | Savings percentage |
|---|---|---|---|---|
| 10 | $10 \times 32 = 320$ | 32 | $320 - 32 = 288$ | 90% |
| 100 | $100 \times 32 = 3{,}200$ | 32 | $3{,}200 - 32 = 3{,}168$ | 99% |
| 500 | $500 \times 32 = 16{,}000$ | 32 | $16{,}000 - 32 = 15{,}968$ | 99.8% |
| 1,000 | $1{,}000 \times 32 = 32{,}000$ | 32 | $32{,}000 - 32 = 31{,}968$ | 99.9% |

**Table 4  Performance comparison.**

| | Data generation time | Complexity of data generation | Storage space | Minimum secure blocks |
|---|---|---|---|---|
| *Kim et al. (2020)* | $T_{ecenc} + 7T_h$ | $O(log_n)$ | $M_{meta} + n \times M_{enc}$ | $3 \times N$ |
| *Liu et al. (2020)* | $2T_{exp} + 2T_{bp} + T_{ReKeyGen}$ | $O(log_n)$ | $M_{meta} + n \times M_{enc}$ | $2 \times N$ |
| *Egala et al. (2021)* | $T_{ecenc} + T_h + T_s$ | $O(n)$ | $M_{meta} + n \times M_{enc}$ | $3 \times N$ |
| Our solution | $n \times T_{exp} + T_{bp} + T_s$ | $O(n)$ | $M_s + n \times M_{enc}$ | $N$ |

requires individual verification, with a complexity of $O(n)$ operations, where nnn is the number of signatures. In our approach, aggregated signatures require a single verification, reducing the complexity to $O(1)$, thus making the system more efficient.

### Performance and storage efficiency comparison

Based on the related work referenced in *Cerchione et al. (2023)*, *Chelladurai & Pandian (2022)*, *Kim et al. (2020)*, *Fatokun, Nag & Sharma (2021)*, *Shuaib et al. (2022)*, *Liu et al. (2020)*, *Guo et al. (2022)*, *Liu et al. (2022)*, *Yuan et al. (2022)*, *Okegbile, Cai & Alfa (2022)*, *Zaabar et al. (2021)*, *Hegde & Maddikunta (2023)*, the proposed scheme is compared with those of *Kim et al. (2020)*, *Liu et al. (2020)*, and *Egala et al. (2021)*. The primary focus of the comparison is to analyze the total computation time, complexity, and storage space required during the authentication, encryption, and signing processes before uploading medical data, as well as the minimum number of blocks required to resist attacks. The results of this comparison are summarized in Table 4.

Assume that $T_{ecenc}$ denotes the encryption time in the elliptic curve cryptography system, $T_h$ represents the computation time of the hash function, $T_{ReKeyGen}$ refers to the re-encryption key generation time, $T_{exp}$ is the time for an exponentiation operation, $T_{bp}$ is the time for a bilinear pairing operation, $T_S$ is the time for generating a signature, and $n$ represents the number of attributes involved in the encryption process.

For storage space, let $M_{meta}$ epresent the storage size of metadata, $M_{enc}$ represent the storage size of the encrypted data, and $M_S$ represent the storage size of the aggregated signature.

Moreover, let $N$ represent the minimum number of blocks required by the HN to prevent an attacker from successfully inferring the data's source or tampering with it, based on attack success probabilities.

Table 4 presents a performance comparison between the proposed scheme and the referenced schemes in terms of data generation time, computational complexity, storage space, and attack resistance (measured by the minimum number of secure blocks). The experimental results indicate that the proposed scheme effectively maintains data generation time within a reasonable range by avoiding re-encryption key generation time $T_{ReKeyGen}$, encryption time in the elliptic curve cryptography system $T_{ecenc}$, and reducing the time for bilinear pairing operations $T_{bp}$. Additionally, the time complexity of the scheme remains at $O(n)$ with no significant increase, ensuring the system's efficiency. By utilizing aggregated signatures $M_s$, the scheme reduces the on-chain data storage burden, thereby optimizing storage efficiency. Moreover, while maintaining a low probability of attack success, the proposed scheme reduces the minimum number of secure blocks required by approximately 2~3 times compared to the referenced schemes, enhancing attack resistance and significantly improving the overall security of the system.

## DISCUSSION

Our findings highlight the effectiveness of a hybrid blockchain-based solution for secure EMR data sharing, optimizing storage, enhancing security, and improving resilience against 51% attacks.

### Comparison with existing literature

Previous studies, such as those by *Kim et al. (2020)*, *Liu et al. (2020)*, and *Egala et al. (2021)*, have explored blockchain-based solutions for EMR data security and decentralized storage, as shown in Table 5. *Kim et al. (2020)* proposed a system based on Elliptic Curve Digital Signature Algorithm, which offers security but is vulnerable to 51% attacks, with high latency and storage overhead. Consequently, their solution is more suitable for small-scale systems. *Liu et al. (2020)* focused on strong key management, but as storage grows with the number of signatures, their method incurs high computational costs and scalability issues, limiting its applicability in large systems. *Egala et al. (2021)* introduced encrypted signatures for stronger forgery resistance and achieved high throughput and low latency. However, their system still faces significant storage overhead, making it less efficient for environments with large data volumes.

In contrast, our tripartite blockchain model uses an aggregate signature scheme that significantly reduces storage overhead and improves scalability, making it highly suitable for large-scale medical data applications. Additionally, it supports low latency and high throughput while addressing the critical issue of 51% attacks, a vulnerability that many existing solutions fail to fully mitigate.

Despite these advantages, our approach has some limitations. Similar to *Kim et al.'s (2020)* model, it requires substantial computational resources to maintain security and performance, which could be challenging in resource-constrained environments. Although the aggregate signature reduces storage overhead, reliance on decentralized storage

**Table 5 Security and efficiency comparison.**

|  | Security | Data privacy protection | Storage efficiency | Application scope |
|---|---|---|---|---|
| *Kim et al. (2020)* | Low resistance to 51% attacks | Basic encryption | High storage overhead ($\approx 2\times$) | Small-scale systems |
| *Liu et al. (2020)* | Moderate resistance | Strong encryption | Storage grows with signatures | Complex applications |
| *Egala et al. (2021)* | Encrypted signatures, strong forgery resistance | Strong privacy (encrypted sig.) | High storage overhead ($\approx 2 \sim 3\times$) | High throughput environments |
| Our solution | Aggregate signatures, strong forgery resistance | Strong privacy (aggregate sig.) | Significant optimization ($\approx 90\%+$) | Large-scale medical data |

systems may introduce potential reliability and performance issues in real-world deployments. Furthermore, while the Proof of Work (PoW) consensus mechanism enhances scalability, it still incurs high computational costs, particularly when handling large volumes of medical data.

### Significance to the research area

Our research advances the field by providing a more secure and efficient method for EMR data sharing. This hybrid blockchain model enhances interoperability between healthcare providers, improving patient care and reducing administrative burdens. It sets a new benchmark for future studies on blockchain-based healthcare systems.

### Broader implications and future research

The principles of our hybrid blockchain approach can be applied to other fields requiring secure data sharing, such as finance and supply chain management. Future research could explore the scalability of our method in larger datasets and real-world implementations, and develop more advanced consensus algorithms and cross-chain protocols.

## CONCLUSION

In conclusion, the hybrid blockchain-based solution proposed in this study provides a robust framework for addressing patient privacy and data security in the smart healthcare domain. By integrating a tripartite blockchain architecture with IPFS technology and a cross-chain signature algorithm, the system ensures efficient data management, secure sharing, and tamper-proof storage of electronic medical records (EMRs). Experimental results demonstrate its effectiveness in mitigating vulnerabilities, such as susceptibility to 51% attacks, and significantly reducing storage overhead while maintaining low latency and high throughput. Moving forward, the system can be further refined through continued research, with potential applications extending beyond healthcare to other sectors, including finance and supply chain management.

## ACKNOWLEDGEMENTS

We would like to express our gratitude to the following authors for their valuable icon contributions used in this work: Icons for Patient blockchain, Healthcare

provider blockchain, Social blockchain, Patient, Hospital, External users, and External network were created by Freepik; the External blockchain network and Level icons were created by Good Ware; the Data Processor and Data processor icons were created by Dewi Sari; the IPFS icons were created by Hilmy Abiyyu A.; the Insurance company icon was created by kerismaker; the Pharmacy icon was created by prettycons; the Government regulatory icon was created by nawicon; the PK&SK icon was created by Creative Stall Premium; the Di and Data icons were created by itim2101; and the External agency icon was created by geotatah. All icons are from www.flaticon.com.

### Funding

This work was supported by the National Natural Science Foundation of China (Grant No. 62102312), Key Research and Development Program of Shaanxi (Program No. 2024GX-YBXM-079), the fund of the State Key Laboratory of Integrated Service Networks (Grant No. ISN24-13), the Young Talent fund of University Association for Science and Technology in Shaanxi, China (Grant No. 20210119), The Youth Innovation Team of Shaanxi Universities (Program No. 23JP160), the Postgraduate Innovation Fund of Xi'an University of Posts and Telecommunications under Grant (Grant No. CXJJDL2024012). No additional external funding was received for this study. The funders had no role in study design, data collection and analysis, decision to publish, or preparation of the manuscript.

### Grant Disclosures

The following grant information was disclosed by the authors:
National Natural Science Foundation of China: 62102312.
Key Research and Development Program of Shaanxi: 2024GX-YBXM-079.
State Key Laboratory of Integrated Service Networks: ISN24-13.
Young Talent fund of University Association for Science and Technology in Shaanxi, China: 20210119.
Youth Innovation Team of Shaanxi Universities: 23JP160.
Postgraduate Innovation Fund of Xi'an University of Posts and Telecommunications: CXJJDL2024012.

### Competing Interests

The authors declare that they have no competing interests.

### Author Contributions

- Gang Han conceived and designed the experiments, performed the computation work, prepared figures and/or tables, and approved the final draft.

- Yan Ma conceived and designed the experiments, performed the experiments, performed the computation work, prepared figures and/or tables, authored or reviewed drafts of the article, and approved the final draft.
- Zhongliang Zhang analyzed the data, authored or reviewed drafts of the article, and approved the final draft.
- Yuxin Wang performed the experiments, analyzed the data, authored or reviewed drafts of the article, and approved the final draft.

## Data Availability

The raw data and code are available in the Supplemental Files.

## Supplemental Information

Supplemental information for this article can be found online at http://dx.doi.org/10.7717/peerj-cs.2653#supplemental-information.

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
