# Peer review of "A hybrid blockchain-based solution for secure sharing of electronic medical record data"

_PeerJ Computer Science, doi:10.7717/peerj-cs.2653_

## Round 0.1 · original submission · Major Revisions

More details are needed in a revision. The novelty needs to be clarified. Proof of concept should be discussed in detail. The related section does not give the current state-of-art. A protocol flow should be added to give the main steps of the proposed solution. A notation table should be provided. In the experimental results section, a comparison should be made by considering implementation results. Table 1 does not make any sense in real-world applications. Figs 5-6-7 do not give a significant contribution and a comparison with currently proposed blockchain-based solutions in terms of experimental results.

Reviewer 1 ·

Basic reporting

The article seems generally appropriate. The use of IPFS and the aforementioned methods such as triple chain and social blockchain have made the study highly theoretical. It does not seem possible to implement and maintain this solution in real life. It may be appropriate to express these restrictions under a heading such as "limitations".

Experimental design

The experimental studies were extremely superficial, raw data was also examined, but the study could be stronger if more details were given.

Validity of the findings

The presented solution seems superior to other studies according to the "External data sharing" and "data fitting" criteria. Are we sure about this comparison and more detail should be given.

Additional comments

The study can be considered original in terms of the method presented. However, the methods used and especially the use of a tool such as IPFS, which is still far from performance and reliability, and the lack of answers to issues such as how PoW verifications will work in real life, make the study seem far from practical application. It may be appropriate to express these restrictions under a heading such as "limitations".

Reviewer 2 ·

Basic reporting

--There are several sections where the text could be clearer. Specifically, please consider revising Lines 23, 77, 121, and 128 for better clarity and comprehension. The manuscript would benefit from a comprehensive language and grammar review to enhance clarity and readability.

--Several sections require restructuring for a coherent flow of ideas. For instance, the introduction and literature review could be more tightly connected to the research question.

Experimental design

--The study's methodology lacks detailed justification for the selection of certain experimental conditions. Clarify why specific choices were made and their relevance to the research question.

--While the research design is suitable, it appears there may be gaps in addressing potential confounding variables. Elaborate on measures taken to mitigate these factors.

--The sample size justification is unclear. Provide a more detailed explanation, including any power analysis conducted to determine the adequacy of the sample size for detecting a meaningful effect.

Validity of the findings

--The analysis section needs significant improvements. Consider revisiting the statistical methods used and ensuring they are appropriate for the data and research questions. Additional details on the analytical procedures will help assess the robustness of the findings.

--Although the findings contribute to the field, the discussion on their broader implications is somewhat limited. Expand on how these results compare with existing literature and their significance to the research area.

Reviewer 3 ·

Basic reporting

The English language should be improved to ensure that an international audience can clearly understand your text, mainly the abstract.

The related work section is not well organized. Authors must try to categorize the papers and present them in a logical way. The authors should explain clearly what the differences are between the prior work and the solution presented in this paper.
The authors should add a table that compares the key characteristics of prior work to highlight their differences and limitations.

Experimental design

The experimental design should be reviewed. the authors have focused only on the MAN threat. the experiments section is too short. it requires more investigation

Validity of the findings

needs to be updated based on the updates of the experimental design

Reviewer 4 ·

Basic reporting

Authors proposed a secure model to share medical data with the help of blockchain. The work addresses the current challenges in the smart healthcare domain. The novelty is evident. Here are a few general points for the authors to further improve the quality of the paper:

Presentation of the Work: The presentation can be enhanced by organizing the content more clearly.
Challenges and Novel Contributions: Include a separate subsection to highlight the challenges in solving the issues and the novel contributions of the work.
Narrative Improvement: The narrative can be improved by providing clear details about all the elements of the proposed model with sub-headings.

Experimental design

Authors are advised to provide more experimental information for each result and explain how the provided results support the objectives of the proposed model. It is recommended that authors include the computational cost results for their model, such as delays and operational costs on the blockchain. Additionally, a comparison of system performance with and without blockchain should be included. Since using blockchain in any system can be complex and costly, it is essential to demonstrate a strong rationale for its use.

Validity of the findings

The security analysis section is not strong. Authors are advised to use security validation tools to demonstrate their system's capabilities. In the comparison section, it is also recommended to compare the proposed model with fortified chain based electronic medical records sharing systems to highlight the novelty of the proposed work.

Additional comments

no comment

---

## Round 0.2 · Major Revisions

In terms of security and efficiency metrics, please provide a comparison between literature solutions and highlight the importance and novelty of your idea. Please extend the discussion section by considering currently proposed solutions. Figures 7-8 need a comprehensive explanation of how these kinds of effects can be obtained and why the proposed solution can provide these results. Please provide proofreading and be careful about writing mistakes.

Reviewer 5 ·

Basic reporting

Basic reporting done

Experimental design

Yes

Validity of the findings

Good

Annotated reviews are not available for download in order to protect the identity of reviewers who chose to remain anonymous.

Reviewer 6 ·

Basic reporting

The authors have addressed the comments/concerns in their revised manuscript.

Experimental design

The authors have addressed the comments/concerns in their revised manuscript.

Validity of the findings

The authors have addressed the comments/concerns in their revised manuscript.

---

## Round 0.3 · Major Revisions

I have identified some remaining some issues on your paper. I added some comments on the pdf paper. I think, the provided suggestions will help to increase the quality of your paper. You should consider the provided comments and add related requirements.

Please ensure that your security analysis and discussion section has been reviewed and edited. Additionally, strengthen the summary and conclusion sections with the experimental results and security analysis results you obtained.

Reviewer 5 ·

Basic reporting

The authors have revised the paper according to the comments. The paper is now
clear and well structured, so I recommend accepting the paper.

Experimental design

good

Validity of the findings

proved

Additional comments

The authors have revised the paper according to the comments. The paper is now
clear and well structured, so I recommend accepting the paper.

---

## Round 0.4 · accepted · Accept

I am happy to announce that the final version is acceptable to PeerJ Computer Science. I consider the current version ready for publication, as all reviewer's comments and editorial revisions have been generally completed. Before the final publication process, please check your manuscript again and ensure there are no grammatical problems or typos.